# Comparative Study of Consequent-Pole Switched-Flux Machines with Different U-Shaped PM Structures

**Ya Li** , **Hui Yang** * and **Heyun Lin**

School of Electrical Engineering, Southeast University, Nanjing 210096, China; seueelab_ly@163.com (Y.L.); hyling@seu.edu.cn (H.L.)
* Correspondence: huiyang@seu.edu.cn; Tel.: +86-152-5186-7159

**Abstract:** This paper presents a comparative study of two consequent-pole switched-flux permanent magnet (CP-SFPM) machines with different U-shaped PM arrangements. In order to address the flux barrier effect in a sandwiched SFPM machine, two different alternate U-shaped PM designs are introduced to improve the torque capability, forming two CP-SFPM machine topologies. In order to reveal the influence of different magnet designs on the torque production, a simplified PM magneto-motive force (MMF)-permeance model is employed to identify the effective working harmonics in the two CP-SFPM machines. The torque contributions of the main working harmonics are subsequently quantified by a hybrid finite-element (FE)/analytical method. Multi-objective genetic algorithm (GA) optimization is then employed to optimize the design parameters of the proposed CP-SFPM machines. In addition, the electromagnetic characteristics of the CP-SFPM machines with two U-shaped PM arrangements are investigated and compared by the FE method. Finally, a 6/13-pole CP-SFPM machine with an optimally selected U-shaped PM structure is manufactured and tested to validate the FE analyses.

**Keywords:** alternate U-shaped permanent magnet (PM); consequent-pole PM (CPM); doubly salient; interior PM (IPM); switched flux

## 1. Introduction

Due to their having high torque/power density and high efficiency, permanent magnet (PM) machines are widely commercialized in electric vehicle (EV) applications [1]. Compared with conventional surface PM (SPM) configuration, the interior PM (IPM) structure exhibits the advantages of convenient PM retaining and better field-weakening capability [2]. Therefore, various IPM topologies, including V-shaped [3], double-layer V-shaped [4], ▽-shaped [5] and multi-layer PM [6] arrangements, have been successfully proposed and extensively investigated in recent years. However, because of the ever-increasing price and unstable supply, the rare-earth material cost of the PM machines is a major concern for EV application [7]. Therefore, in order to address above-mentioned issue, it is necessary to improve the PM utilization of electrical machine design.

In order to further improve the PM utilization ratio, various spoke-type PM topologies with the flux concentrated effect are extensively employed in dual-stator [8,9], outer rotor [10] and ferrite magnet configurations [11–15]. Compared with single stator topology, a dual-stator spoke-type machine exhibits higher a power factor and torque density [8]. Moreover, the modular outer rotor in-wheel machine with spoke-type PM arrangement is proved to be able to provide higher air-gap flux density and better field-weakening capability than its SPM counterpart [10]. To decrease the cost, low-cost ferrite magnets are adopted in spoke-type configurations for electric vehicle applications [11–14]. It shows that the cost competitiveness can be significantly improved by employing low-cost ferrite magnets. However, due to low remanence flux density and intrinsic coercivity, ferrite magnets generally suffer from a potentially irreversible demagnetization risk [11–14]. The performance comparison of the optimal designed IPM machines with either NdFeB or

ferrite magnets is reported in [15]. It demonstrates that the ferrite magnet machines exhibit comparable efficiency and 40% cost reduction compared to the NdFeB PM cases.

However, due to the flux leakage and rotor rib issues of the conventional spoke-type PM structures [16–18], an airspace barrier design [18] is proposed to simplify the rotor manufacturing without sacrificing the torque capability. Moreover, for the spoke-type vernier PM machines, it is difficult for low-order working harmonics to pass through PMs to form closed loops, namely, the flux barrier effect [19]. This will reduce effective low-order harmonics and the torque capability. Similarly to [18], the alternate flux barriers are utilized in a consequent pole spoke-PM vernier machine [20,21], which exhibits approximately 57% higher torque density than its conventional counterpart [22]. Moreover, compared with conventional spoke-PM vernier machines, the alternate V- and U-shaped PM topologies achieve 80% torque improvement together with a comparable power factor [23]. Recently, the flux bridge design concept was extended to a switched-flux PM (SFPM) machine, which provides additional magnetic paths for the main working harmonic fields [24–26]. As a result, the torque capability can be significantly improved compared with conventional SFPM machines.

In order to address the flux barrier effect in sandwiched SFPM machines [27–30], a consequent-pole SFPM (CP-SFPM) machine with an alternate U-shaped PM design was developed [31–33], which provides an effective circulating path for low-order field harmonics. As a result, the torque capability can be further improved compared with the conventional cases [33]. In order to reveal and evaluate the influence of the magnet type on the electromagnetic performance, two CP-SFPM machines with different alternate U-shaped PM designs are comparatively analyzed in this paper, which also identifies their different features and torque production mechanisms. The two machine topologies and analytical models with different U-shaped PM structures are introduced in Section 2. Furthermore, a hybrid finite element (FE)/analytical model is employed to quantify the torque contributions of multiple working harmonics. In Section 3, the multi-objective genetic algorithm (GA) optimization is then utilized to optimize the design parameters of the two machines in order to obtain a satisfactory torque quality. In Section 4, the electromagnetic characteristics of the two proposed machines are investigated and compared by the FE method in order to select the optimal PM design. In Section 5, the experimental measurement of a 6/13-stator/rotor-pole CP-SFPM machine with better overall performance is carried out to validate the FE analyses. Finally, a comprehensive conclusion is drawn in Section 6.

## 2. Machine Topologies and Analysis Models

### 2.1. Machine Topologies

The configurations of two 6-stator-slot/13-rotor-pole CP-SFPM machines with different U-shaped PM arrangements are illustrated in Figure 1. It shows that both the CP-SFPM machines utilize alternate U-shaped PM configurations to provide an effective circulating path for low-order field harmonics, which indicates that the torque capability can be further improved. The two CP-SFPM machines are regarded as model-I and model-II, respectively; the only difference of which is the position of the radially magnetized PMs, as illustrated in Figure 1. In order to perform a fair comparison, the two machines share the same active stack length, outer stator diameter, air-gap length and current density, etc. In addition, it should be noted that the iron bridges are adopted in the two CP-SFPM machines to strengthen the stator mechanical stability.

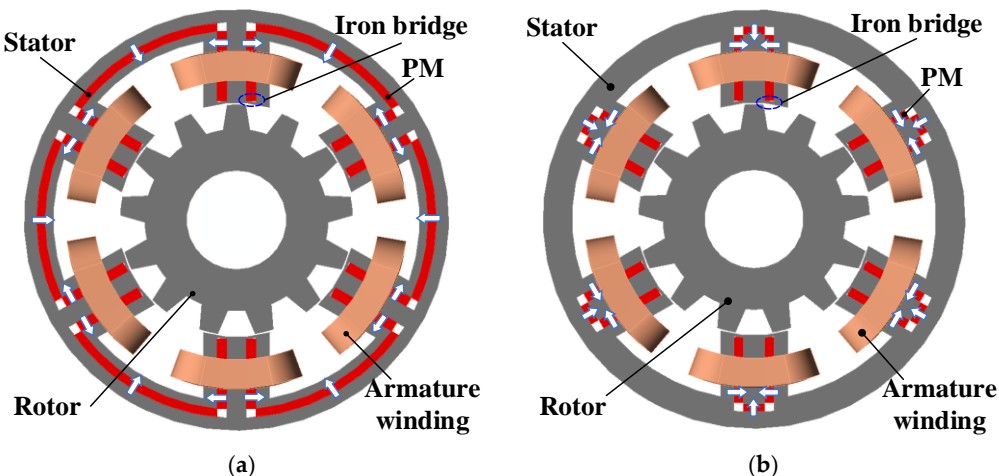

**Figure 1.** Topologies of the 6-stator-slot/13-rotor-pole consequent-pole switched-flux permanent magnet (CP-SFPM) machines with different U-shaped permanent magnet (PM) arrangements. (**a**) Model-I. (**b**) Model-II.

### 2.2. Analytical Modelling

For a double salient structure, the air-gap permeance can be obtained by considering stator and rotor slotting effects, respectively. The air-gap permeance due to the stator slotting can be expressed as [34]

$$\Lambda_s(\theta_s) = \Lambda_{s0} + \sum_{m=1}^{\infty} \Lambda_{sm} \cos(mZ_s\theta_s) \tag{1}$$

where $Z_s$ is the number of stator slots, $\Lambda_{s0}$ and $\Lambda_{sm}$ are the 0th and $m$th components of the air-gap permeance due to the stator slotting, respectively. Similarly, when considering the rotor salient, the air-gap permeance due to the rotor slotting can be expressed as

$$\Lambda_r(\theta_s, t) = \Lambda_{r0} + \sum_{n=1} \Lambda_{rn} \cos[nZ_r(\theta_s - \theta_0 - \Omega_r t)] \tag{2}$$

where $Z_r$ is the number of rotor teeth, $\Omega_r$ is the angular velocity. $\theta_s$ is the rotor position reference to stator. $\theta_0$ is angle of initial rotor position. $\Lambda_{r0}$ and $\Lambda_{rn}$ are the 0th and $n$th components of the air-gap permeance due to the rotor slotting, respectively. Thus, the air-gap permeance function can be obtained as [34,35]

$$\Lambda_g(\theta_s, t) = \frac{\Lambda_s \Lambda_r}{\mu_0/g} = \frac{g}{\mu_0}\Lambda_{r0}\Lambda_{s0} + \frac{g}{\mu_0}\Lambda_{r0}\sum_{m=1}\Lambda_{sm}\cos(mZ_s\theta_s) +$$
$$\frac{g}{\mu_0}\Lambda_{s0}\sum_{n=1}\Lambda_{ri}\cos[nZ_r(\theta_s - \theta_0 - \Omega_r t)] +$$
$$\frac{g}{2\mu_0}\sum_{m=1}\sum_{n=1}\Lambda_{rn}\Lambda_{sm}\cos[(mZ_s \pm nZ_r)\theta_s \mp nZ_r(\Omega_r t + \theta_0)] \tag{3}$$

When $\theta_s = 0$, i.e., for the axis of phase A windings, the ideal PM magneto-motive-force (MMF) waveforms of the two machines are illustrated in Figure 2. $\theta_1$ is the half of the inner stator teeth arc.

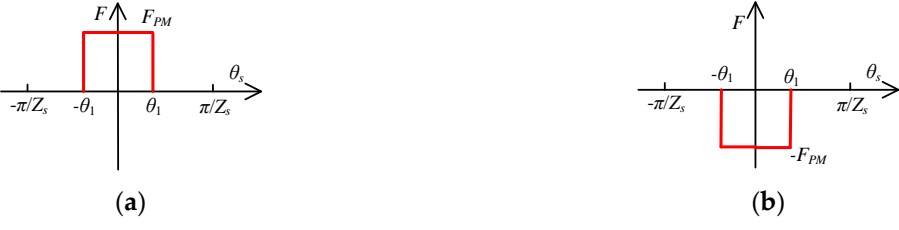

**Figure 2.** Ideal PM magneto-motive-force (MMF) waveforms. (**a**) Model-I. (**b**) Model-II.

It can be seen that the two CP-SFPM machines can be regarded as consequent pole PM configurations, the PM MMF of which can be expressed as [35]

$$F_{PM}(\theta_s) = \sum_{i=0} F_i \sin(iZ_s\theta_s) \tag{4}$$

where $F_i$ is the $i$th PM MMF coefficient. In order to confirm the above-mentioned analyses, the actual PM MMF distributions can be obtained by the FE method [36], as illustrated in Figure 3. It can be seen that the PM MMF waveforms of the CP-SFPM machines are asymmetrical, which indicates that the biased flux effect occurs [37], i.e., 0th harmonic. In addition, compared with model-I, model-II exhibits higher 0th and $iZ_s$th ($i$ = 1, 2, 3, 6) harmonic magnitudes, which is mainly due to its better flux concentrated effect. As a result, relatively larger working harmonic amplitudes and higher torque capability can be achieved in model-II.

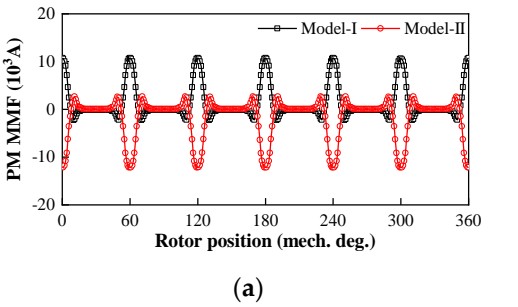

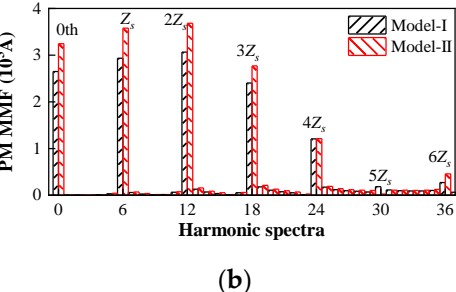

**Figure 3.** Actual PM MMF distributions of the two CP-SFPM machines. (**a**) Waveforms. (**b**) Harmonic spectra.

The air-gap flux density can be obtained by

$$B_g(\theta_s, t) = F_{PM}(\theta_s)\Lambda_g(\theta_s, t) \tag{5}$$

Because the stationary air-gap flux density harmonic components are absent for torque production, the air-gap flux density can be further rewritten as

$$B_g(\theta_s, t) = \sum_j B_{gj}(\theta_s, t) \tag{6}$$

The four components of the air-gap flux density can be, respectively, expressed as

$$B_{g1}(\theta_s, t) = \frac{g}{\mu_0}\Lambda_{s0}F_0\sum_{n=1}^{\infty}\Lambda_{rn}\cos[nZ_r(\theta_s - \theta_0 - \Omega_r t)] \tag{7}$$

$$B_{g2}(\theta_s, t) = \frac{g}{2\mu_0}F_0\sum_{m=1}^{\infty}\sum_{n=1}^{\infty}\Lambda_{rn}\Lambda_{sm}\cos[(mZ_s \pm nZ_r)\theta_s \mp nZ_r(\Omega_r t + \theta_0)] \tag{8}$$

$$B_{g3}(\theta_s, t) = \frac{g}{2\mu_0}\Lambda_{s0}\sum_{n=1}^{\infty}\sum_{i=0}^{\infty}\Lambda_{rn}F_i\sin[(iZ_s \pm nZ_r)\theta_s - nZ_r(\theta_0 - \Omega_r t)] \tag{9}$$

$$B_{g4}(\theta_s, t) = \frac{g}{4\mu_0} \sum_{n=1}^{\infty} \sum_{m=1}^{\infty} \sum_{i=0}^{\infty} \Lambda_{rn} \Lambda_{sm} F_i \sin[(mZ_s \pm nZ_r \pm iZ_s)\theta_s \mp nZ_r(\Omega_r t + \theta_0)] \tag{10}$$

Furthermore, the flux linkage of phase A can be expressed as

$$\psi_A = r_g l_{stk} \int_0^{2\pi} B_g(\theta_s, t) N_a(\theta_s) d\theta_s \tag{11}$$

where $r_g$ is the air-gap radius, $l_{stk}$ is the active stack length. $N_a(\theta_s)$ is the winding function, which can be expressed as

$$N_a(\theta_s) = \sum_{j=1} \frac{2}{j\pi} N_s k_{wj} \cos(j\theta_s) \tag{12}$$

where $N_s$ is number of the series-connected winding turns per phase, $k_{wj}$ is the winding factor of the $j$th air-gap flux density. Thus, the back electromotive force (EMF) of phase A can be obtained by

$$e_A(t) = -\frac{d\psi_A}{dt} \tag{13}$$

The electromagnetic torque can be expressed as

$$T_e = \frac{e_A i_A + e_B i_B + e_C i_C}{\Omega_r} \tag{14}$$

where $e_A$, $e_B$ and $e_C$ are the back-EMFs of phases A, B and C, respectively. $i_A$, $i_B$, and $i_C$ are the current of phases A, B and C, respectively. Because the fractional-slot concentrated windings are adopted in CP-SFPM machines, the reluctance torque is very low and can be neglected [36]. Thus, the average torque can be further rewritten as

$$T_{avg} = 3r_g l_{stk} N_s I_A Z_r \Lambda_{r1} \times$$
$$\left\{ \frac{g}{\mu_0} \frac{1}{Z_r} \Lambda_{s0} F_0 k_{w|Z_r|} + \frac{g}{2\mu_0} F_0 \sum_{m=1}^{\infty} \frac{\Lambda_{sm} k_{w|mZ_s \pm Z_r|}}{mZ_s \pm Z_r} + \frac{g}{2\mu_0} \Lambda_{s0} \sum_{i=1}^{\infty} \frac{F_i k_{w|iZ_s \pm Z_r|}}{iZ_s \pm Z_r} + \frac{g}{4\mu_0} \sum_{m=1}^{\infty} \sum_{i=1}^{\infty} \frac{\Lambda_{sm} F_i k_{w|mZ_s \pm Z_r \pm iZ_s|}}{mZ_s \pm Z_r \pm iZ_s} \right\} \tag{15}$$

where $I_A$ is the peak value of the phase current. According to (15), it can be observed that only rotor fundamental permeance, all PM MMF and stator permeance harmonic components are responsible for the effective torque production, which is similar to the other stator PM machines [38,39].

Considering the main working harmonics, the average torque can be further rewritten as

$$T_{avg} = \sum_{j=1} T_{avgj} = 3r_g l_{stk} N_s I_A \times$$
$$\left[ \begin{array}{l} \frac{Z_r}{|Z_s - Z_r|} B_{|Z_s - Z_r|} k_{w|Z_s - Z_r|} + \frac{Z_r}{|Z_s + Z_r|} B_{|Z_s + Z_r|} k_{w|Z_s + Z_r|} + \frac{Z_r}{|2Z_s - Z_r|} B_{|2Z_s - Z_r|} k_{w|2Z_s - Z_r|} + \\ \frac{Z_r}{|2Z_s + Z_r|} B_{|2Z_s + Z_r|} k_{w|2Z_s + Z_r|} + \frac{Z_r}{|3Z_s - Z_r|} B_{|3Z_s - Z_r|} k_{w|3Z_s - Z_r|} + \frac{Z_r}{|3Z_s + Z_r|} B_{|3Z_s + Z_r|} k_{w|3Z_s + Z_r|} \end{array} \right] \tag{16}$$
$$= 3r_g l_{stk} N_s I_A \sum_{j=1} \frac{Z_r}{P_j} B_j k_{wj}$$
$$= 3r_g l_{stk} N_s I_A \sum_{j=1} G_{rj} B_j k_{wj}$$

where $B_j$, $k_{wj}$ and $P_j$ are the $j$th air-gap flux density magnitude, winding factor, and pole pairs, respectively. $T_{avgj}$ is the average torque generated by the $j$th air-gap flux density. $G_{rj}$ is the gear ratio of the $j$th field harmonics, which is defined as the ratio of the $Z_r$ to the $P_j$. According to (16), it can be observed that the torque capability can be significantly improved by low-order harmonics due to its amplification effect of the $G_{rj}$. In this case ($Z_s = 6$, $Z_r = 13$), those working harmonics with orders of "$|iZ_s - Z_r|$ (1st)" refer to low-order ones, e.g., 1 ($i = 2$) and 5 ($i = 3$). Thus, in order to enhance low-order field harmonics, two alternate U-shaped PM configurations are proposed in this paper.

*2.3. Torque Quantification by a Hybrid FE/Analytical Approach*

In order to reveal the torque production mechanism in a more intuitive way, a hybrid FE/analytical approach was developed in order to quantify the torque components of the main-order working harmonics. The analytical procedure is shown in Figure 4. Because the average torque components of the main-order working harmonics cannot be quantified directly by FE method, the hybrid FE/analytical approach was adopted in this paper. Moreover, compared with the analytical method, the hybrid FE/analytical approach provides a more accurate prediction of the air-gap flux density distributions. As a result, an exact torque quantification of the two CP-SFPM machines can be obtained by employing the proposed hybrid approach. It can be seen that the torque contributions of the various working harmonics can be quantified by unified torque equation when the no-load air-gap flux density is obtained by employing a static magnetic field simulation, which indicates that the computational time can be significantly reduced compared with transient simulation. The torque proportion of the *j*th air-gap flux density can be defined as

$$\lambda_j = \frac{T_{avgj}}{T_{avg}} \times 100\% \tag{17}$$

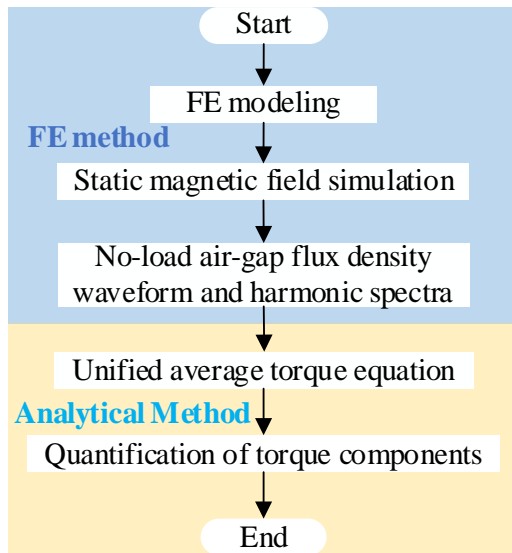

**Figure 4.** The procedure of the proposed hybrid finite element (FE)/analytical approach.

The corresponding actual open-circuit air-gap flux density waveforms and their harmonic spectra are shown in Figure 5. It shows that model-II has higher $|2Z_s - Z_r|$ (1st) order harmonic magnitude, but lower $|Z_s - Z_r|$ (7th), $|3Z_s - Z_r|$ (5th) and $|Z_s + Z_r|$ (19th) order harmonic amplitudes. The open-circuit flux density distributions of the two machines are illustrated in Figure 6. It can be seen that both the CP-SFPM machines exhibit high flux density in stator yoke, which indicates that larger low-order effective harmonics and better torque capability can be achieved in the proposed designs. In addition, the paths of $|2Z_s - Z_r|$ (1st) harmonic are highlighted with blue lines in Figure 6, which clearly suggests that model-II provides a more effective circulating path for low-order field harmonic, as evidenced in Figure 5b, which is mainly attributed to better flux concentrated effect and relative lower magnetic saturation in stator yoke for model-II case.

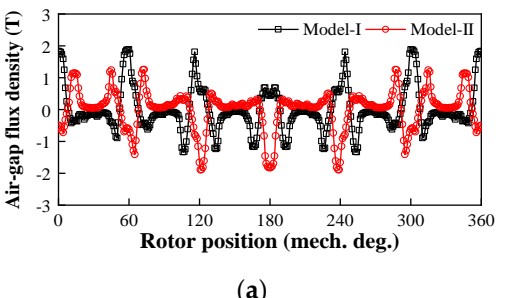 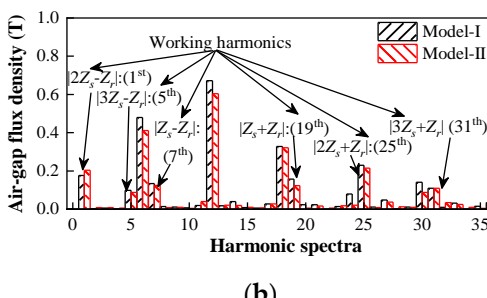

(**a**)　　　　　　　　　　　　(**b**)

**Figure 5.** Open-circuit air-gap flux density. (**a**) Waveforms. (**b**) Harmonic spectra.

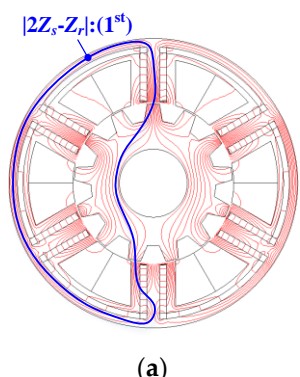 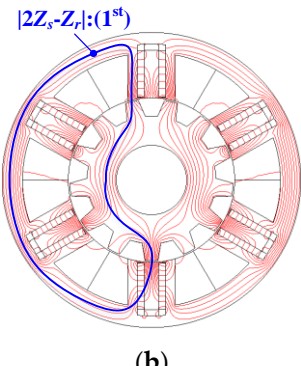

(**a**)　　　　　　　　　　　　(**b**)

**Figure 6.** Open-circuit magnetic field distributions. (**a**) Model-I. (**b**) Model-II.

In order to confirm the above-mentioned theoretical analyses, the torque contributions of various working harmonics in the two CP-SFPM machines are quantified by (16), as illustrated in Figure 7a. It can be found that the low-order harmonic, i.e., $|2Z_s - Z_r|$ (1st) order, contributes the highest average torque in the two machines, which confirms that the U-shaped PM design provides a relatively lower reluctance for low-order harmonics, and hence torque capability is improved. According to (17), the torque proportions of various harmonics are shown in Figure 7b. Due to the higher magnitude of the $|2Z_s - Z_r|$ (1st) order air-gap flux density, as shown in Figure 5b, model-II exhibits higher torque proportion of the $|2Z_s - Z_r|$ (1st) order harmonic compared with model-I, as illustrated in Figure 7b, which indicates that higher torque can be obtained in model-II. The detailed torque contributions of the main working harmonics are tabulated in Table 1. Because only the main working harmonics are taken into consideration, the total average torques calculated by the foregoing equations are slightly lower than FE predictions, as shown in Table 1.

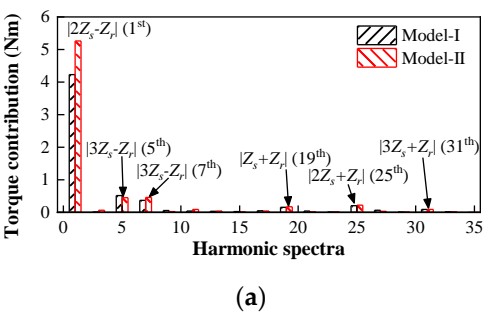 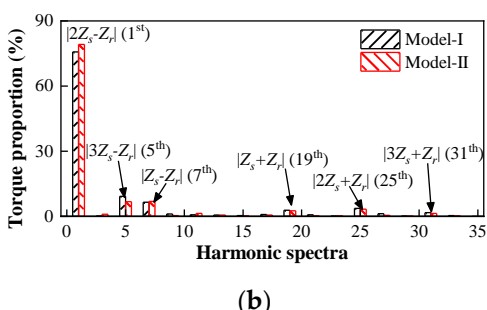

(**a**)　　　　　　　　　　　　(**b**)

**Figure 7.** Torque contributions and torque proportion of the main working harmonics in the two CP-SFPM machines. (**a**) Torque contributions. (**b**) Torque proportions.

**Table 1.** Torque Contributions of the Main Working Harmonics of the Two Proposed CP-SFPM Machines.

| Pole Pairs | Model-I | Model-II |
|---|---|---|
| | Harmonics Magnitude (T)/Torque Contribution (Nm)/Torque Proportion (%) | |
| $\lvert 2Z_s - Z_r \rvert$, 1st | 0.167/4.216/75.420 | 0.204/5.274/79.189 |
| $\lvert 3Z_s - Z_r \rvert$, 5th | 0.102/0.515/9.213 | 0.087/0.450/6.757 |
| $\lvert Z_s - Z_r \rvert$, 7th | 0.102/0.368/6.583 | 0.124/0.457/6.862 |
| $\lvert Z_s + Z_r \rvert$, 19th | 0.117/0.155/2.773 | 0.124/0.168/2.523 |
| $\lvert 2Z_s + Z_r \rvert$, 25th | 0.201/0.203/3.631 | 0.215/0.222/3.333 |
| $\lvert 3Z_s + Z_r \rvert$, 31th | 0.114/0.141/2.522 | 0.110/0.091/1.367 |
| Total torque | Sum (%)/FE-predicted (Nm) | |
| | 5.59/5.66 | 6.66/6.74 |

## 3. Design Optimization

### 3.1. Rotor Pole Selection

By taking 6-stator-slot structure as an example, the torque characteristics of the two proposed CP-SFPM machines with different rotor pole numbers are given in Figure 8 in order to select a feasible rotor pole number. It should be noted that the model-II structure exhibits higher torque capability than the model-I case in all selected rotor pole numbers, as shown in Figure 8a, which is mainly due to the better flux concentrated effect and hence larger effective air-gap field harmonics in model-II design. Taking average torque and torque ripple into account, the 13-pole rotor structure was selected for the following analysis.

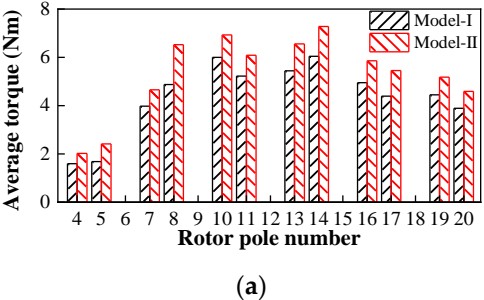
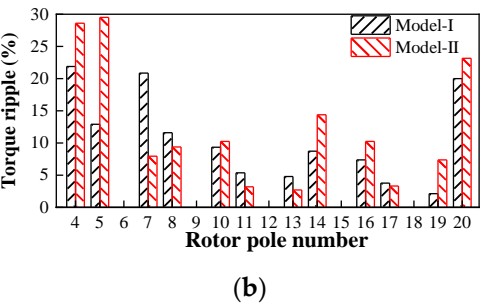

(**a**)  (**b**)

**Figure 8.** Torque quality of 6-stator-slot CP-SFPM machines with different rotor pole numbers. (**a**) Average torque versus rotor pole number. (**b**) Torque ripple versus rotor pole number.

### 3.2. Multi-Objective Optimization

In order to perform a fair comparison, the multi-objective GA optimization method embedded in the JMAG17.1 software package was employed to optimize the torque quality of the two machines. The optimization objectives are to maximize the average torque and minimize the torque ripple, of which the weight factors are set as 1 and 0.5, respectively. The number of generations and population size are set as 100 and 20, respectively. Furthermore, all the design parameters, as shown in Figure 9, are taken into consideration. The geometric parameters are globally optimized with the restriction of the preceding optimized PM dimensions, so as to maximize the torque capability at the flux-enhanced state. The scatter diagrams regarding the average torque versus the torque ripple are given in Figure 10. In order to balance high average torque and low torque ripple, the optimal Pareto front curves of the two CP-SFPM machines with blue lines are illustrated in Figure 10. The detail design parameters of the selected cases for the two CP-SFPM machines are tabulated in Table 2.

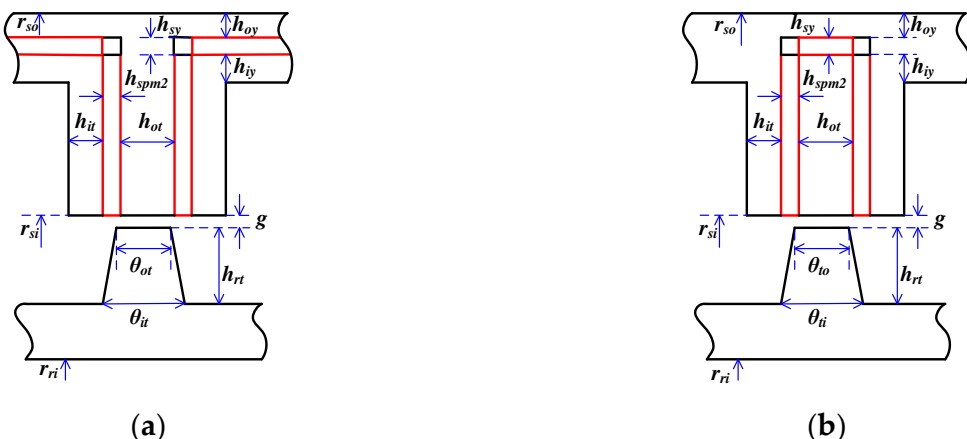

**Figure 9.** Illustration of design parameters of CP-SFPM machines with different PM arrangements. (**a**) Model-I. (**b**) Model-II.

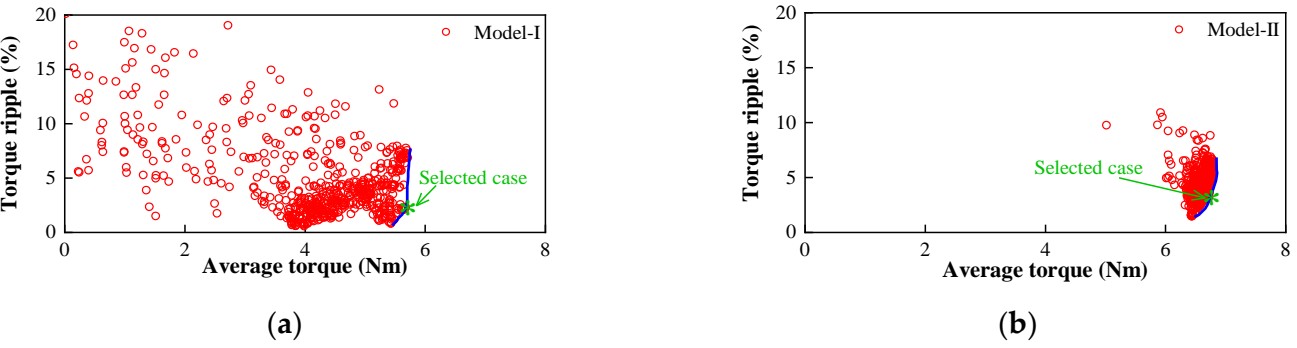

**Figure 10.** Scatter diagrams of the average torque versus the torque ripple. (**a**) Model-I. (**b**) Model-II.

**Table 2.** Main Design Parameters of the Two Proposed CP-SFPM Machines.

| Parameters | Model-I | Model-II |
|---|:---:|:---:|
| Stator slot number, $Z_s$ | 6 | |
| Rotor pole pairs number, $Z_r$ | 13 | |
| Stator outer diameter, $r_{so}$, mm | 102 | |
| Stator inner diameter, $r_{si}$, mm | 56.8 | |
| Stator yoke PM thickness, $h_{sy}$, mm | 2 | |
| Stator inner teeth thickness, $h_{it}$, mm | 2.3 | |
| Stator outer teeth thickness, $h_{ot}$, mm | 3.2 | |
| Stator PM thickness, $h_{spm}$, mm | 2.4 | |
| Iron bridge thickness, $h_{ib}$, mm | 0.5 | |
| Active stack length, $l_{stk}$, mm | 50 | |
| Air-gap length, $g$, mm | 0.5 | |
| Rotor outer diameter, $r_{ro}$, mm | 55.8 | |
| Rotor teeth outer arc, $\theta_{to}$, deg | 9.2 | |
| Rotor teeth inner arc, $\theta_{ti}$, deg | 16.9 | |
| Rotor teeth length, $h_{rt}$, mm | 5.9 | |
| Rotor inner diameter, $r_{ri}$, mm | 24 | |
| PM grade | N42SH | |
| PM volume, mL | 43.37 | 25.96 |
| Steel grade | 35CS250 | |
| Turns per phase | 130 | |
| Rated current, A | 8 | |

## 4. Electromagnetic Characteristics Investigation

In order to confirm the optimal U-shaped PM structure for the proposed CP-SFPM machines, the basic electromagnetic characteristics are comparatively investigated in this Section.

### 4.1. Open-Circuit Performance

The open-circuit back-EMFs are given in Figure 11. It can be seen that the model-II configuration exhibits a 24.21% higher fundamental component than the model-I arrangement, which is mainly attributed to the higher $|2Z_s - Z_r|$ (1st) order harmonic in the model-II configuration. The cogging torque waveforms are illustrated in Figure 12. It can be seen that the peak cogging torque of the model-I configuration is higher than the model-II arrangement, which is mainly attributed to the existence of more abundant air gap flux field harmonics in the model-I case, as evidenced in Figure 5.

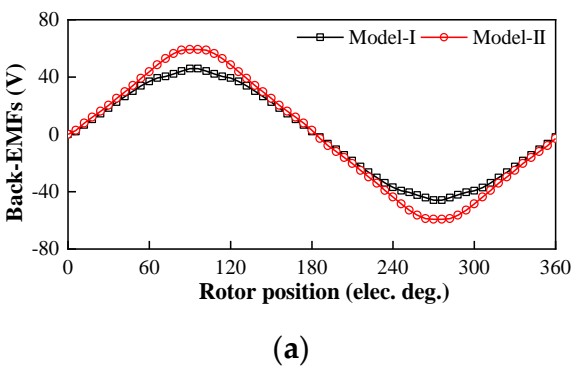
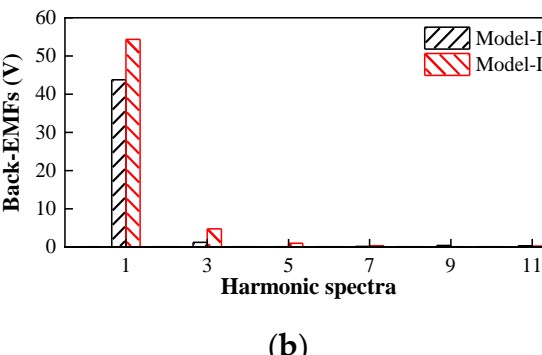

(**a**)  (**b**)

**Figure 11.** Back-electromotive force (EMF) waveforms @1000 r/min. (**a**) Waveforms. (**b**) Harmonic spectra.

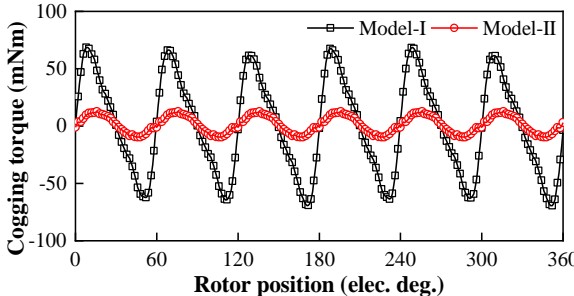

**Figure 12.** Cogging torque waveforms.

### 4.2. On-Load Torque Characteristics

The torque characteristics of the two CP-SFPM machines are illustrated in Figure 13. It can be seen that the maximum torques of the two machines are reached at approximately 10 and 15 electrical degrees, respectively. Due to its higher $|2Z_s - Z_r|$ (1st) air-gap flux density harmonic, the model-II arrangement exhibits 19.26% higher torque than the model-I type configuration, as shown in Figure 13b. In addition, the model-II configuration exhibits better overload capability regardless of the load current, as evidenced in Figure 13c. The average torque/PM volume against the phase current curves is given in Figure 13d. It can be observed that the model-II arrangement exhibits higher PM utilization for torque generation, which indicates that the total cost of PM can be significantly reduced by adopting the model-II configuration.

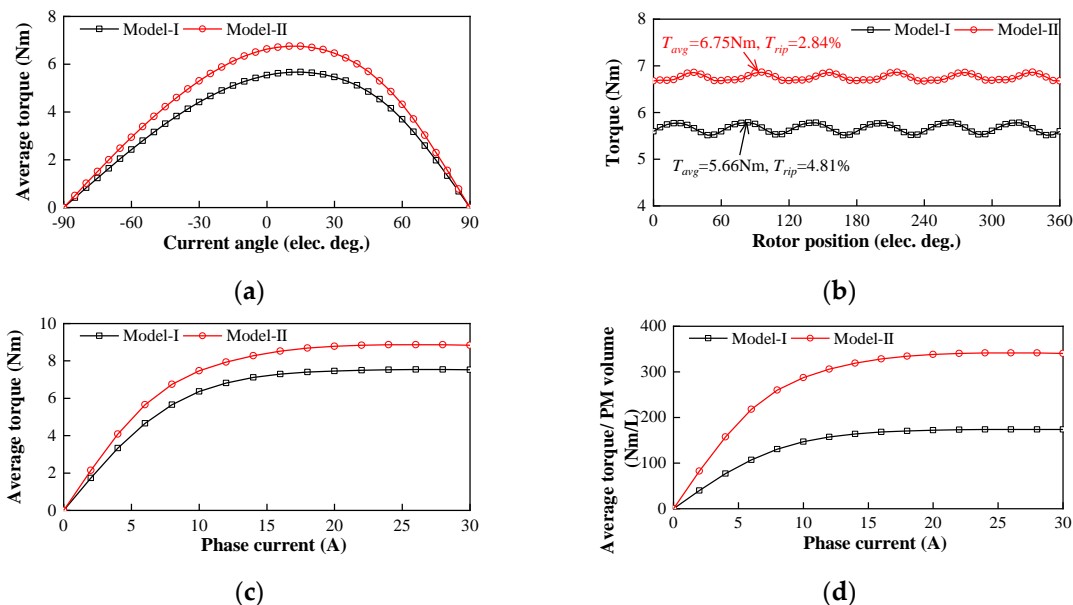

**Figure 13.** Torque characteristics. (**a**) Average torque versus current angle. (**b**) Steady torque. (**c**) Average torque versus phase current. (**d**) Average torque/PM volume versus phase current.

### 4.3. Loss and Efficiency Characteristics

The iron loss versus speed curves of the two CP-SFPM machines at rated current condition are given in Figure 14a. It can be seen that the iron losses of the two machines increase with rotor speed. In addition, due to its higher air gap flux field harmonics, the model-I configuration has larger iron loss than the model-II structure regardless of rotor speed. The PM eddy-current losses against speed curves are given in Figure 14b. The model-I design has higher PM eddy-current loss than the model-II case, which is mainly due to the larger reluctance for armature-reaction flux and hence higher PM current density. The corresponding main losses of the two studied CP-SFPM machines are illustrated in Table 3. The efficiency maps of the two machines are illustrated in Figure 15. It can be observed that the maximum efficiencies of the two CP-SFPM machines are 90.03% and 91.67%, respectively. Due to its relatively lower iron loss and PM eddy-current loss, the model-II configuration exhibits a larger high efficiency region.

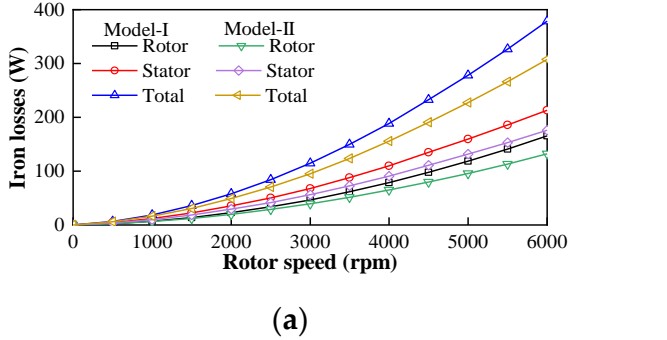

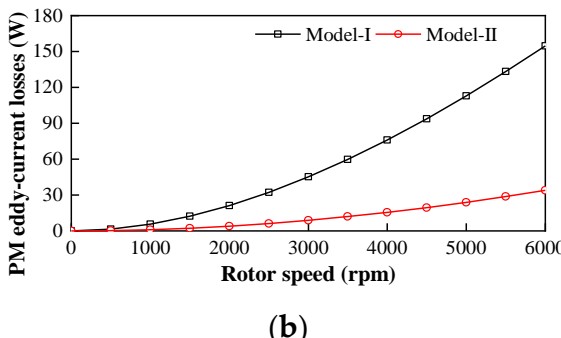

**Figure 14.** Loss characteristics. (**a**) Rated-load iron losses against speed curves. (**b**) PM eddy-current losses against speed at rated-load.

**Table 3.** Main Losses of the Two Proposed CP-SFPM Machines.

| Items | | CP-SFPM Machines | |
|---|---|---|---|
| | | **Model-I** | **Model-II** |
| Iron loss, W | rotor | 8.93 | 8.48 |
| | stator | 14.87 | 13.69 |
| PM eddy-current loss, W | | 5.63 | 1.01 |
| Copper loss, W | | 43.46 | |
| Total loss, W | | 72.89 | 66.65 |
| Maximum efficiency, (%) | | 90.03 | 91.67 |

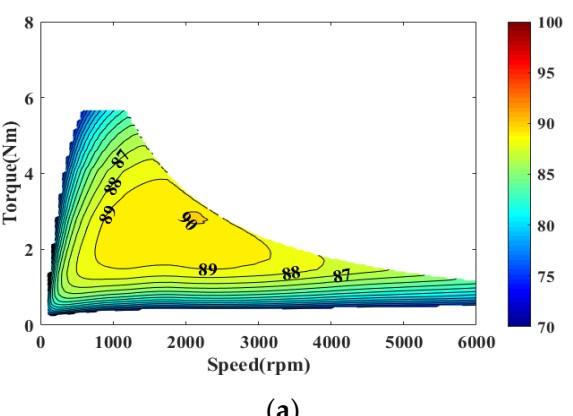

(**a**)

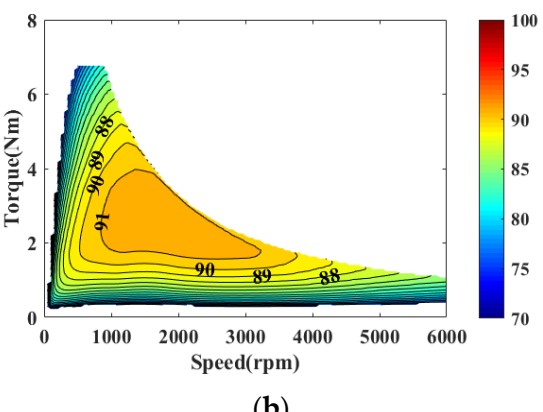

(**b**)

**Figure 15.** Efficiency maps. (**a**) Model-I. (**b**) Model-II.

## 5. Experimental Validation

In order to validate the aforementioned FE analyses, a 6/13-stator/rotor pole model-II was manufactured. The stator and rotor assemblies are illustrated in Figure 16a,b, respectively. The open-circuit 3D flux density distributions of the CP-SFPM machine with U-shaped PM arrangement are shown in Figure 17. The axial leakage flux in 3D FE model can be clearly observed, which results in higher mismatches of 2D FE-predicted back-EMF with the measurements, as illustrated in Figure 18. The measured, hybrid FE/analytical and FE-predicted average torque against phase current curves are given in Figure 19. It should be noted that the measured torque values are slightly lower than the FE-predicted ones, which is mainly due to the end leakage and mechanical tolerance. Moreover, in order to present a clear comparison, the average torques at different phase currents predicted by the 2D/3D FE, hybrid FE/analytical and measured methods are given in Table 4. Because the hybrid FE/analytical average torques are based on the 2D FE-predicted air-gap flux density, these values are more closed to the 2D FE predictions. However, because the main-order working harmonics are taken into consideration, the hybrid FE/analytical average torques are slightly lower than 2D FE predictions, which confirms the effectiveness of proposed hybrid approach. Moreover, the 3D FE-predicted average torques are closer to the measured values, which is mainly due to the fact that the end leakage is taken into consideration in 3D FE simulation. Furthermore, the measured and FE-predicted torque/power–speed curves are shown in Figure 20. Similarly, a slightly smaller mismatch between the 3D FE and measured values can be observed, as illustrated in Figure 20, which is mainly due to the fact that the end effect is taken into consideration in 3D FE prediction. Overall, the measured results of back-EMF and average torque agree well with the FE-predicted results, which validates the FE analyses.

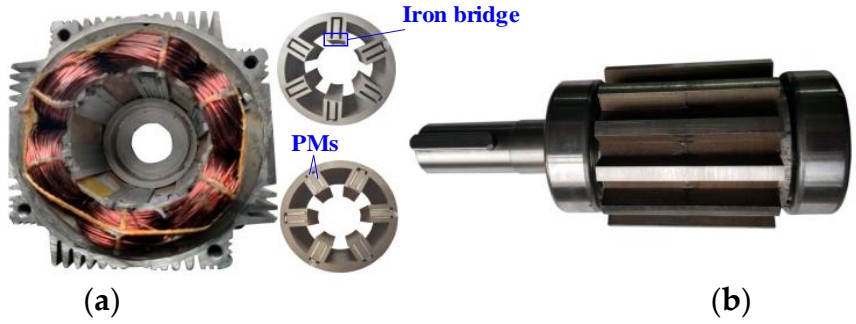

**Figure 16.** The prototype of the CP-SFPM machine with model-II structure. (**a**) Stator. (**b**) Rotor.

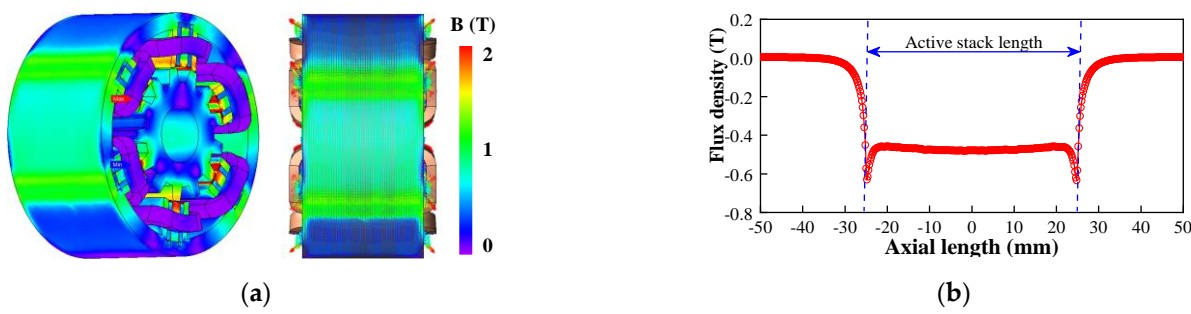

**Figure 17.** Open-circuit flux density distributions of the model-II CP-SFPM machine. (**a**) 3D flux density distributions. (**b**) Air-gap flux density waveform in the axial direction at zero rotor position.

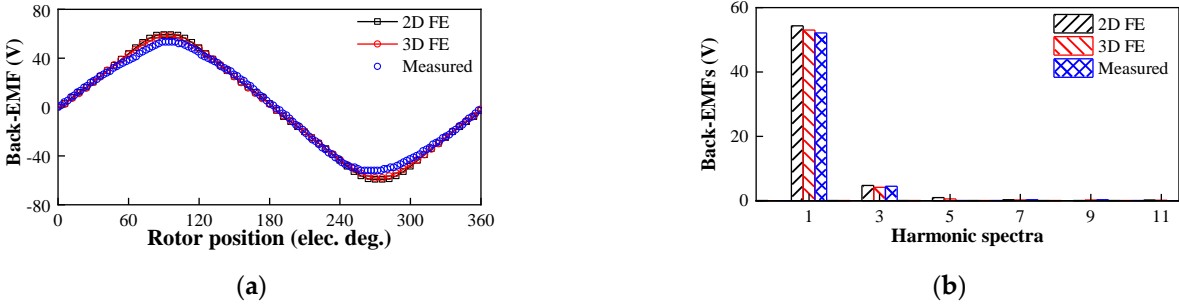

**Figure 18.** Comparison of FE-predicted and measured back-EMFs. (**a**) Waveforms. (**b**) Harmonic spectra.

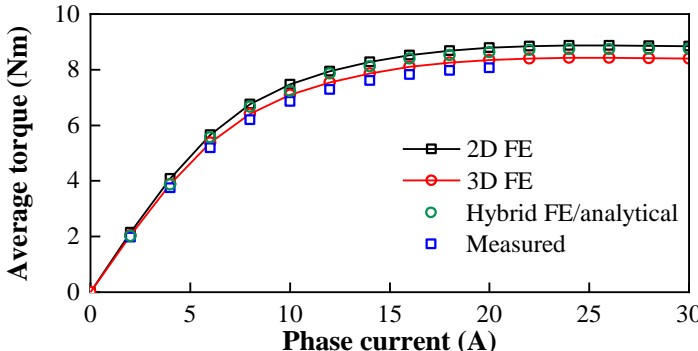

**Figure 19.** Measured and FE-predicted average torque against phase current.

**Table 4.** Average torques of the CP-SFPM model-II at different phase currents.

| Items | CP-SFPM Machines | | |
|---|---|---|---|
| | $I_{rms}$ = 4 A | $I_{rms}$ = 8 A | $I_{rms}$ = 12 A |
| 2D FE, (Nm) | 4.09 | 6.75 | 7.94 |
| 3D FE, (Nm) | 3.89 | 6.42 | 7.54 |
| Hybrid FE/analytical, (Nm) | 3.87 | 6.66 | 7.86 |
| Measured, (Nm) | 3.75 | 6.20 | 7.29 |

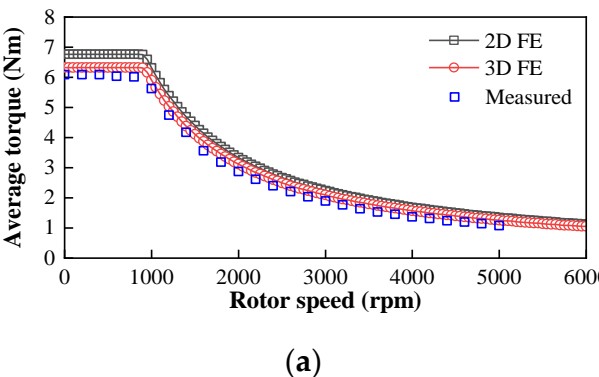

(**a**)

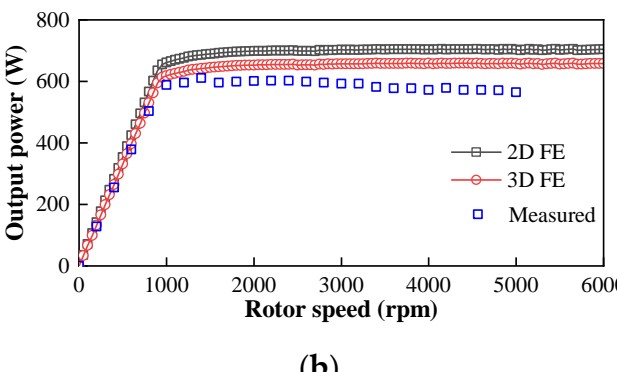

(**b**)

**Figure 20.** Comparison of FE-predicted and measured average torque/power–speed curves. (**a**) Torque–speed curves. (**b**) Power–speed curves. ($U_{dc}$ = 150 V).

## 6. Conclusions

This paper presents a comparative study of two CP-SFPM machines with different U-shaped PM arrangements. There are some conclusions can be summarized such as:

1.  The performance comparison of two CP-SFPM machines with different U-shaped magnets was carried out, which provides a guidance design for this type of machine.
2.  A hybrid FE/analytical model for elaborating the torque production mechanism of the investigated machines, as well as comprehensive comparative analyses, are presented, which can clearly identify the dominant air-gap field harmonics contributing to the torque production, as well as reveal the underlying reason why model-II topology can deliver the higher torque capability of the two U-shaped PM designs.
3.  The electromagnetic characteristics of the two optimized CP-SFPM machines were comparatively investigated. It can be found that model-II exhibits s higher torque capability and PM utilization ratio, which are mainly attributed to higher low-order working harmonics. Moreover, model-II has a wider high-efficiency region than the model-I case, which is mainly due to its relatively lower iron loss and PM eddy-current loss. Therefore, the CP-SFPM machine with model-II structure exhibits a more potential practical EV application than the model-I case.

Finally, some experimental measurements on a 6/13-pole CP-SFPM machine prototype with the model-II structure validate the theoretical and FE analyses.

**Author Contributions:** Conceptualization, Y.L., H.Y. and H.L.; methodology, Y.L., H.Y. and H.L.; software, Y.L.; validation, Y.L.; formal analysis, Y.L.; writing—original draft preparation, Y.L.; writing—review and editing, Y.L., H.Y. and H.L. and funding acquisition, H.Y. and H.L.; All authors have read and agreed to the published version of the manuscript.

**Funding:** This work was jointly supported in part by National Natural Science Foundations of China under Grant (52037002 and 52077033) in part by Natural Science Foundation of Jiangsu Province for Youth (BK20170674), in part by the Fundamental Research Funds for the Central Universities (2242017K41003), in part by "Hong Kong Scholar" Program (XJ2018014), in part by the "SEU Zhishan Young Scholars" Program of Southeast University (2242019R40042), in part by the Scientific Research Foundation of Graduate School of Southeast University (YBPY1974), in part by Jiangsu Planned

Projects for Postdoctoral Research Funds (1116000266), and in part by "The Excellence Project Funds of Southeast University".

**Conflicts of Interest:** The authors declare no conflict of interest.

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
