# Peer review of "Comparative Study of Consequent-Pole Switched-Flux Machines with Different U-Shaped PM Structures"

_wevj, doi:10.3390/wevj12010022_

Round 1

Reviewer 1 Report

This paper proposes an interesting comparative study supported by FE simulations and experimental validations.

  1. The main issue with this proposal is the vehicle context that is totally absent. In this context, the authors should clearly show what makes the proposed comparative study interesting to EV propulsion drivetrain. The introduction section should be therefore re-written accordingly while the reference section should be updated.
  2. The contribution should accordingly be better highlighted.
  3. The design optimization is obscure as there are no details on the used GAs. Did the authors just use software?
  4. The experimental validation section is weak and should be enhanced with more critical discussions.

Author Response

Dear Reviewer 1,

We would like to sincerely thank you for spending precious time on the manuscript and the reviewers for their constructive comments and suggestions, which are fully taken into account in our revision.

The modifications are detailed as follows and highlighted in RED in the revised paper.

Yours sincerely

The authors

Reviewer 2 Report

This paper presents a comparative study of two consequent-pole switched-flux permanent magnet (CP-SFPM) machines with different U-shaped PM arrangements. Detailed analysis method, simulation and experimental results are given. It is a good paper. Here are some comments.

  1. Section 2.3 investigates a hybrid FE/analytical approach for torque modelling. More details should be given to show why this kind of hybrid approach is required. What are the advantages of approach compared with analytical approach or FE approach?
  2. Section 3.2 investigates the multi-objective optimization of two CP-SFPM machines. Optimal Pareto fronts are given in Figure 10. However, the detailed information of the optimization, including the optimization model, like objectives and constraints, optimization parameters, and optimization methods are not given.
  3. Regarding the parameters, as shown in Figure 9, there are many parameters. Are they considered in the optimization?
  4. There are many optimal points in Figure 10. Which one is selected for the performance comparison?
  5. Figure 19 shows a comparison of the motor performance by using three methods, 2D, 3D, and measurement. Why the hybrid approach is not considered? Also, to have a clear comparison, some values should be given to show the error of different methods.

Author Response

Dear Reviewer 2,

We would like to sincerely thank you for spending precious time on the manuscript and the reviewers for their constructive comments and suggestions, which are fully taken into account in our revision.

The modifications are detailed as follows and highlighted in RED in the revised paper.

Yours sincerely

The authors

Round 2

Reviewer 1 Report

The authors have properly addressed the reviewer raised issues and given comments.

Reviewer 2 Report

The revision can be accepted for publication. A suggestion: a detailed response should be given in the revision process as it clearly shows the comments that have been responded to.